# A scalable Bayesian continual learning framework for online and sequential decision making

**Hanwen Xing**
Nuffield Department of Women's
Reproductive Health
University of Oxford
`hanwen.xing@wrh.ox.ac.uk`

**Christopher Yau**
Nuffield Department of Women's
Reproductive Health
University of Oxford
HDRUK
`christopher.yau@wrh.ox.ac.uk`

## Abstract

Continual learning (CL) refers to the ability to continually learn and exploit new knowledge while retaining experiences accumulated from past. Though numerous CL methods have been proposed in recent years, it is not straightforward to deployed them directly to online or sequential decision making problems due to the computational burden and the lack of uncertainty quantification. In this paper, we focus on Instance-incremental classification problems with concept shift, and propose a online/sequential decision making model based on a novel scalable Bayesian continual learning framework that provides i) statistically principled and computationally efficient Bayesian knowledge updating scheme and ii) scalable and exact posterior inference procedure based on a Mixture of Experts model. In addition, as an exemplar-free method, our method does not require storing or modelling any previously seen instances, making it appealing to e.g. online decision making problems in biomedical applications where data privacy is of concern.

## 1 Introduction

Continual learning (CL) enables an intelligent system to develop and refine itself adaptively in accordance with real-world dynamics by incrementally accumulating and exploiting knowledge gained from past experience without the need to train any new model from scratch [Hassabis et al., 2017]. The main challenge CL has to tackle is known as *catastrophic forgetting*, which refers to previously learned knowledge being drastically interfered by new information [McClelland et al., 1995, McCloskey and Cohen, 1989]. In order to deliver accurate and trustworthy predictions, a CL model in practice should, on one hand, be able to integrate new knowledge and update existing knowledge efficiently based on the stream of new inputs from dynamic data distributions (learning plasticity) and, on the other hand, maximally retain relevant information from the past and prevent catastrophic forgetting (memory stability). The competition between these two conflicting objectives is known as the *stability-plasticity dilemma*, which has been widely studied from both biological and computational perspectives [Ditzler et al., 2015, Parisi et al., 2019]. See also Wang et al. [2024] for a comprehensive review. The adaptive and sequential nature of CL makes it appealing for online and sequential decision making problems, where the decision boundary may vary temporally or spatially, and one needs to accurately identify and efficiently adapt to such changes. In this paper, we propose a sequential decision making model based on a novel scalable Bayesian CL framework.

Following the notation in Wang et al. [2024], we denote $\mathcal{D}_{t,b} = \{\mathcal{X}_{t,b}, \mathcal{Y}_{t,b}\}$ the $b$th batch of samples of a task $t$, where $t, b \in \mathbb{N}$ are the task and batch index respectively, $\mathcal{X}_{t,b}$ is the feature data and $\mathcal{Y}_{t,b}$ is the data labels. Here a task refers to a distinct learning problem. In this paper, we focus on the

Workshop on Bayesian Decision-making and Uncertainty, 38th Conference on Neural Information Processing Systems (NeurIPS 2024).

Instance-incremental classification problem under a *concept shift* [Kurle et al., 2019] scenario: we assume i) training samples are from a single task $t = 1$ and come in batches, ii) data labels $\mathcal{Y}_b$ are categorical and iii) the distribution of features $p_{t,b}(\mathcal{X}_{t,b})$ remain constant over all batches $b$ while the conditional $p_{t,b}(\mathcal{Y}_{t,b}|\mathcal{X}_{t,b})$ could vary with $b$ due to underlying change points. This scenario is common in practice: For example, in the context of biomedical application, the health risks of a group of individuals with reasonably stable physical characteristics such as height or weight are still likely to vary due to external factors such as weather change or pandemic outbreak. The goal of the proposed CL-based sequential decision making model is to estimate this temporally varying decision boundary without retaining previously seen data or training a new model from scratch.

Existing state-of-the-art CL approaches such as Chaudhry et al. [2019], Kumari et al. [2022] and Wu et al. [2018] are computationally intensive, and require either estimating a full generative model of the input data or storing past examples as part of their memory states, which are infeasible in many applications due to privacy or storage concerns. Exemplar-free CL [He and Zhu, 2022] and regularization-based methods such as Kirkpatrick et al. [2017] and Li and Hoiem [2017] avoid the need of storing or modelling input data, but they are also computational costly and lack a statistically principled updating scheme or uncerntainty quantification. These undesirable traits make them unsuitable for problems such as online or sequential decision making. To address these issues, we propose a sequential decision making model based on a conceptually simple and scalable Bayesian continual learning framework that combines statistically principled modelling, and flexible machine learning methods. Our proposed approach utilizes scalable online exact Bayesian update and ensembles learning. The exact Bayesian updating framework ensures resilience to catastrophic forgetting, improves robustness of the model, and gives predictions with easy-to-interpret uncertainty quantification, which is crucial in decision making process. On the other hand, the ensemble framework provides extra flexibility to cope with non-stationarity: by aggregating individual base models that are aware of the potential distributional difference in training samples, the model can adapt to it more efficiently by choosing the best combination of past experiences using both evidence from data and user's domain knowledge.

## 2 Method

Our proposed framework consists of two components: a fixed pre-trained feature-extractor and an ensemble of base models. We start by introducing the base model.

### 2.1 Base Gaussian process classifier

Each base model in our proposed framework is an online Gaussian process classifier. To achieve exact and computationally efficient prediction, we use the Dirichlet-based GP classification [Milios et al., 2018], a fast and accurate GP-based model that interprets categorical labels as samples from a Dirichlet distribution, and combine it with WISKI-GP [Stanton et al., 2021], a scalable online Gaussian process model based on structured kernel interpolation [Wilson and Nickisch, 2015], to ensure that the base model can be updated tractably and efficiently. This modelling choice ensures both efficient and exact posterior inference, and scalable sequential update *without storing or reusing any previous samples*: Information extracted from past data is represented by and updated through a interpolating kernel [Wilson and Nickisch, 2015] whose size only depends on the number of inducing points. Let $c$ be the Dirichlet distribution hyperparameter, $\nu$ be the WISKI-GP kernel hyperparameter, $p_{c,\nu}(\cdot|X_{new}, \mathcal{X}_{prev}, \mathcal{Y}_{prev})$ be the predictive distribution of label associated with a new data point $X_{new}$ conditioned on historical observations $\{\mathcal{X}_{prev}, \mathcal{Y}_{prev}\}$. Let $p_{c,\nu}(\mathcal{X}_{prev}, \mathcal{Y}_{prev})$ be the marginal likelihood of $\{\mathcal{X}_{prev}, \mathcal{Y}_{prev}\}$. In this work, we follow the suggestion in Milios et al. [2018], set the Dirichlet hyperparameter $c = 0.1$.

### 2.2 Feature extractor

Similar to Petit et al. [2023], we incorporate a feature extractor into our learning framework. We train a feature extractor (e.g. parameteric UMAP [Sainburg et al., 2021] or VAE [Kingma and Welling, 2013]) based on a collection of samples $\{\mathcal{X}_{init}, \mathcal{Y}_{init}\}$ prior to the continual learning process. This $\{\mathcal{X}_{init}, \mathcal{Y}_{init}\}$ can either be a historical dataset or simply the first batch $\{\mathcal{X}_1, \mathcal{Y}_1\}$. Once it has been trained, the feature extractor is fixed throughout the learning process as in Prabhu et al. [2023], and the inference and update steps are based on the embedded features generated by the feature extractor.

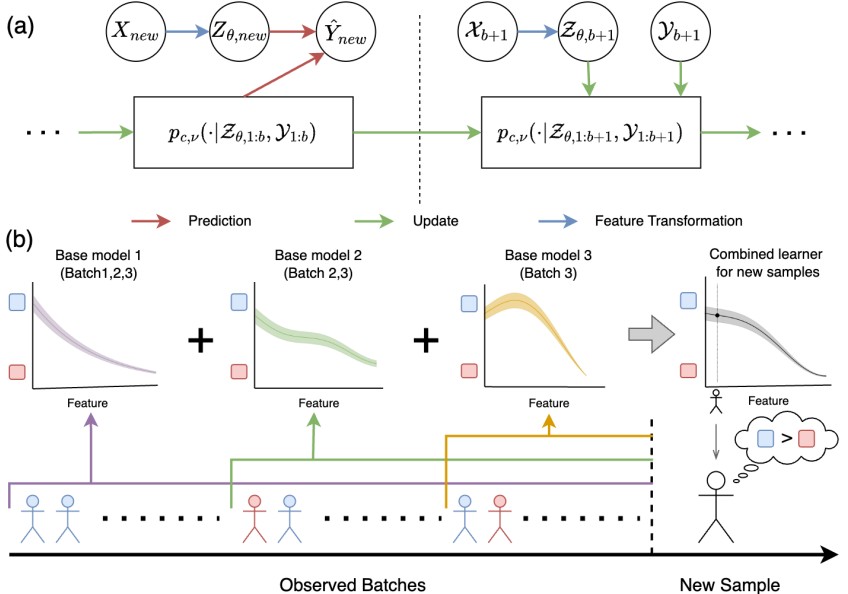

Figure 1: **(a)**: Graphical illustration of the prediction and update scheme for a base learner. **(b)**: A schematic illustration of the ensemble of three base models given three batches using a binary classification example.

## 2.3 Ensemble learning for Instance-incremental update

Here we discuss the ensemble learning procedure. We omit the task index subscript $t$ for brevity. Suppose the feature extractor $P_\theta$ parameterized by $\theta$ and the WISKI-GP hyperparameter $\nu$ have been chosen, and we denote $\mathcal{Z}_{\theta,1:b}$ the transformed feature obtained by applying $P_\theta$ to $\mathcal{X}_{1:b}$, and $Z_{\theta,new}$ the transformed feature of a generic test point $X_{new}$. Under a static setup where both $p_b(\mathcal{X}_b)$ and $p_b(\mathcal{Y}_b|\mathcal{X}_b)$ remain constant, updating the posterior predictor $p_{c,\nu}(\cdot|Z_{\theta,new}, \mathcal{Z}_{\theta,1:b}, \mathcal{Y}_{1:b})$ w.r.t. $b$ sequentially using Bayes rule is straightforward and computationally efficient thanks to the caching strategy given by Stanton et al. [2021]. However, under a non-stationary setup, the standard Bayesian updating scheme may not be flexible enough to adapt to the potential change points, as Bayes rule implies that all historical samples have same contribution to the predictor.

We propose an ensemble learning framework to address this issue. Whenever a new batch $b \geq 2$ arrives, we first initialise a new base predictor $p_{c,\nu_b}(\cdot|Z_{\theta,new}, \mathcal{Z}_{\theta,b}, \mathcal{Y}_b)$ based solely on the $b$th batch where $\nu_b$ is the associated WISKI-GP hyperparameter, and then update the $b-1$ previous base predictors $p_{c,\nu_{b'}}(\cdot|Z_{\theta,new}, \mathcal{Z}_{\theta,b':b}, \mathcal{Y}_{b':b})$, $b' = 1, ..., b-1$ using Bayes rule. Finally the $b$ base predictors are combined to give the ensemble predictor at step $b$:

$$p_{ens,b}(\cdot|Z_{\theta,new}) = \sum_{b'=1}^{b} W_{b'} p_{c,\nu_{b'}}(\cdot|Z_{\theta,new}, \mathcal{Z}_{\theta,b':b}, \mathcal{Y}_{b':b}) \tag{1}$$

where the weights $W_{b'} \propto p_{c,\nu_{b'}}(\mathcal{Z}_{\theta,b}, \mathcal{Y}_b|\mathcal{Z}_{\theta,b':b-1}, \mathcal{Y}_{b':b-1})$ for $b' = 1, ..., b-1$, $W_b \propto p_{c,\nu_b}(\mathcal{Z}_{\theta,b}, \mathcal{Y}_b)$ and $\sum_{b'=1}^{b} W_{b'} = 1$. The marginal likelihoods $p_{c,\nu_{b'}}(\mathcal{Z}_{\theta,b}, \mathcal{Y}_b|\mathcal{Z}_{\theta,b':b-1}, \mathcal{Y}_{b':b-1})$ can also be computed efficiently using the caching procedure given by Stanton et al. [2021]. Under the assumption that test samples and the most recently seen data $\mathcal{D}_t$ follow the same distribution, this weighting strategy allows the model to make predictions on test samples by utilising previous knowledge in an automatic and adaptive fashion: when $\mathcal{D}_t$ aligns well with the previous knowledge, the ensemble would favour base models that contain more historical experience as those models provide more informative (i.e. concentrated) priors that are inline with the likelihood of $\mathcal{D}_t$. On the other hand, when there is a prior-data conflict between the historical experience and the newly observed data $\mathcal{D}_t$ due to e.g. change points, the ensemble would down weight the contribution from historical experience, and make predictions primarily based on the more recently observed data batches. See also Fig 1 for a schematic illustration. Thanks to the grid-interpolating kernel used in WISKI-GP, each base model in the ensemble is essentially characterized by a fix-sized $M \times M$

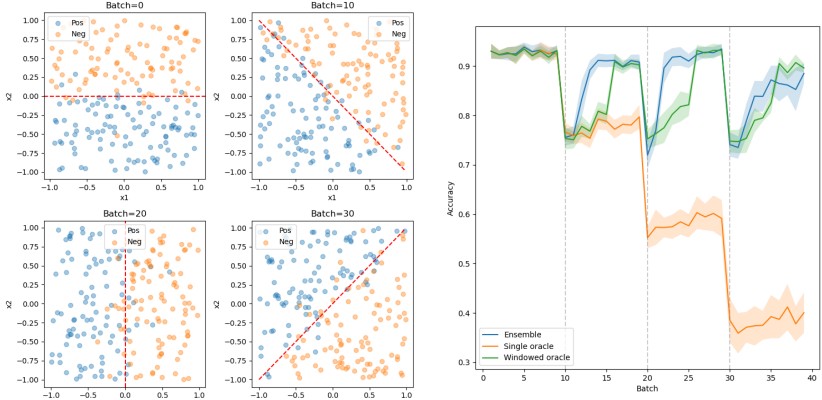

Figure 2: **Left**: Training data in batch 0, 10, 20 and 30. **Right**: One-batch-ahead prediction accuracy through out the learning process. Results are averaged over 10 independent runs. Shaded region corresponds to the $2\sigma$ error band of the averaged accuracy. Grey vertical lines indicate change points.

kernel matrix where $M$ is the number of inducing points. As a result, even though the ensemble needs to store an increasing number of models as new batches come in, it is manageable in most of the scenarios.

## 3 Numerical examples

We demonstrate our method using a modified version of the rotating-boundary example in Kurle et al. [2019]: Let $T = 40$, $B = 200$ be the number of batches and the sample size of each batch. Let data $\mathcal{D}_t = \{\mathcal{X}_t, \mathcal{Y}_t\}$ for $t = 1, ..., T$ where $\mathcal{X}_t = \{X_{t,i}\}_{i=1}^B$, $X_{t,i} \sim \text{Uniform}([-1,1]^2)$ and $\mathcal{Y}_t = \{Y_{t,i}\}_{i=1}^B$, $Y_{t,i} \sim \text{Bernoulli}(\sigma(w_t^T X_{t,i}))$ for $i = 1, ..., B$. Here $\sigma(\cdot)$ is the sigmoid function and $w_t = [10\sin(\lfloor t/10 \rfloor \pi/4), 10\cos(\lfloor t/10 \rfloor \pi/4)]$ is a time-dependent parameter that rotates the decision boundary clockwise by $\pi/4$ clockwise every 10 batches, altering the conditional distribution of labels given the feature vector (Fig 2). Since in this example the feature vectors are low-dimensional, we use the raw features directly.

We compare the prediction accuracy of our proposed ensemble method with 1) a single oracle GP that has access to all previously seen data and gives prediction based on the full history and 2) a single oracle GP that has access to at most 5 most recently observed batches (i.e. a memory window of length 5) and gives prediction conditioned on the observed batches within the memory window. We stress that unlike the oracle GPs, our ensemble does not require storing or reusing any previous data. We compare the one-batch-ahead prediction accuracy given by different models and report the results in Fig 2. We find that our proposed method is able adapt to change points faster than both the single and windowed oracle GPs without accessing any previously seen data, indicating the efficacy of our proposed approach.

## 4 Conclusion

We propose an online/sequential decision making model based on a scalable and conceptually simple Bayesian continual learning framework under the *concept drift* setup. Our method combines scalable online exact Bayesian update and ensembles learning. Comparing with existing CL methods, our approach is computationally efficient and statistically more principled thanks to the Bayesian framework, and does not require storing or reusing previously seen samples, making it appealing in applications where data privacy is of concern. In this work, we only consider the scenario where the conditional distribution of label changes over batches (temporally varying decision boundary). We also plan to extend our approach to the scenario where both the conditional distribution of labels and the feature distribution change over batches using techniques such as indirect discriminant alignment [Liu et al., 2020].

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
