# OpenReview forum: "A scalable Bayesian continual learning framework for online and sequential decision making"
_NeurIPS.cc/2024/Workshop/BDU — NeurIPS BDU Workshop 2024 Poster_

### Official Review · Reviewer_L5eX · 2024-09-24
**Elegant and simple approach to continual learning via ensemble / Bayesian model averaging**

**Rating:** 6
**Confidence:** 4

**Review:**

# Summary

This paper provides a simple and elegant solution to continual learning under the assumption that data are provided in batches.
The core idea is that a separate, efficient probabilistic model is built for each batch (in this case, the model is a scalable and efficient Gaussian process classification representation). In particular, it is important that the (predictive, marginal) likelihood can be evaluated efficiently for each model. Upon arrival of a new batch, the weights for each previous batch (=model) are updated using Bayes rule (à la Bayesian model averaging). Prediction for a new batch / data point is then performed using these weights and the previous models.

# Evaluation

To my knowledge, while the Bayesian approach to continual learning is very common, the paper's idea of considering each batch predictor as a separate model and then effectively applying Bayesian model averaging is novel and original. This approach is significant in that this is a well-known open problem and the proposed method can seemingly apply to any domain in which we can efficiently build and update a probabilistic model that summarizes each batch. The quality is acceptable for a paper workshop -- the idea is good but the empirical evaluation is extremely limited to a small toy model. The paper is generally clear, but it seems it has been written hastily (several typos, a few sentences whose English could be improved).

# Comments

A few comments and perhaps limitations worth exploring or stating better in a future version of the paper:

1. It seems the method as proposed will critically depend on the quality of the feature extractor. Basing the extractor entirely on the first batch, as suggested, seems dubious? Regardless of how the extractor is trained, since it is fixed, there are concerns whether the extractor will still deal successfully with concept drift. I understand that the authors assume $p(X)$ is stationary, but my intuition is that the feature extractor might struggle if it sees $(x, y)$ combinations never encountered before, especially in higher dimension; it'd be good to see some empirical exploration. The $2D$ example in the paper does not use a feature extractor.
2. While it is true that the method does not store individual data points, it still stores a summary of each batch since it needs to keep one model per batch (in this case, the summary is represented by the scalable GP posterior). So the overall storage is still $O(N)$, under the reasonable assumption that the number of batches is proportional to the number of data points. The "summary" models might be quite large. Still, I see that storing summaries is better than storing individual data points.
3. The paper hinges on the idea that data arrives in batches, which may or may not be the case. Of course, any sequence of streaming data can be arbitrarily divided into batches, but then there is the question of how big each batch should be. The authors might want to explore this and give maybe some heuristic recommendations (if not a fully principled solution).

# Conclusions

Overall, this is a nice paper with an interesting idea that deserves further exploration and discussion, and very suitable in terms of topic for the workshop, and this is what I am mostly using as a base for my final rating. I don't give a higher score because the execution and demonstration is very limited, even for a workshop submission. I am looking forward to a full paper version with more extensive experiments.

---

### Official Review · Reviewer_sxzj · 2024-09-27
**Review of Bayesian CL**

**Rating:** 5
**Confidence:** 3

**Review:**

**Summary:**
The paper proposes a Bayesian continual learning model for classification with concept shift. The authors show that the proposal is computationally more scalable to existing work, and does not need to store samples.

**Strength:**
- The model is well motivated and tackles an important problem.
- The intuition of the proposed model is clearly described.
- The numerical results (though limited) illustrates the advantage of the proposed method in terms of robustness and adaptivity to data shift.

**Weakness:**
- Some of the model details are missing or not well explained. For example, the choice of the ensemble weight is not justified properly
- The Bayesian update of the predictors is not clearly stated. The readers could benefit more if Section 2.3 can provide more technical details.
- The numerical example seems overly simplified without the feature extraction stage (since it's a 2-dimensional problem). It is unclear how the proposed method perform in more complex scenarios.
- The numerical study is also somewhat limited. For example, oracles of different window size should at least be compared instead of only comparing the full history and a somewhat arbitrary case of 5-batch window.

---

### Decision · Program_Chairs · 2024-10-09

Accept (Poster)